# The Role of Centrosome Distal Appendage Proteins (DAPs) in Nephronophthisis and Ciliogenesis

**DOI:** 10.3390/ijms222212253

**Published:** 2021-11-12

**Authors:** Fatma Mansour, Felix J. Boivin, Iman B. Shaheed, Markus Schueler, Kai M. Schmidt-Ott

**Affiliations:** 1Department of Nephrology and Medical Intensive Care, Charité-Universitätsmedizin Berlin, 10117 Berlin, Germany; fatma-sayed-mansour@charite.de (F.M.); Felix.boivin@mdc-berlin.de (F.J.B.); 2Molecular and Translational Kidney Research, Max-Delbrück-Center for Molecular Medicine in the Helmholtz Association (MDC), 13125 Berlin, Germany; 3Department of Pathology, Faculty of Veterinary Medicine, Cairo University, 12613 Giza, Egypt; imanshaheed@yahoo.com

**Keywords:** distal appendages, centrosome, ciliogenesis, transition fibers, nephronophthisis

## Abstract

The primary cilium is found in most mammalian cells and plays a functional role in tissue homeostasis and organ development by modulating key signaling pathways. Ciliopathies are a group of genetically heterogeneous disorders resulting from defects in cilia development and function. Patients with ciliopathic disorders exhibit a range of phenotypes that include nephronophthisis (NPHP), a progressive tubulointerstitial kidney disease that commonly results in end-stage renal disease (ESRD). In recent years, distal appendages (DAPs), which radially project from the distal end of the mother centriole, have been shown to play a vital role in primary ciliary vesicle docking and the initiation of ciliogenesis. Mutations in the genes encoding these proteins can result in either a complete loss of the primary cilium, abnormal ciliary formation, or defective ciliary signaling. DAPs deficiency in humans or mice commonly results in NPHP. In this review, we outline recent advances in our understanding of the molecular functions of DAPs and how they participate in nephronophthisis development.

## 1. Introduction

The cilium is a thin evolutionarily conserved microtubule-based organelle that protrudes from the apical surface of most mammalian cell types into the extracellular space. Although once considered to be vestigial organelles, cilia have been found to profoundly influence tissue development and homeostasis [1,2,3]. Since studies of the intraflagellar transport in the green alga Chlamydomonas in the 1990s and the discovery that cilia are linked to renal diseases [4,5], much has been learned about the role of the cilium as a conduit for signal transduction pathways. Due to its receptor-enriched membrane, the primary cilium integrates and modulates numerous signaling pathways that are critical for vertebrate development and organ differentiation [2,6,7,8,9], including the Hedgehog (Hh) [10], Wingless (Wnt) [11], mammalian target of Rapamycin (mTOR), G protein-coupled receptors (GPCR) [12], platelet-derived growth factor receptor (PDGFR)-alpha [13], and transforming growth factor (TGF)-beta and Notch pathway [14]. Moreover, cilia play important roles in planar cell polarity and in cell cycle regulation [1,2]. Despite our improved understanding about the relevance of cilia and an ever-increasing level of their biological functions, the role of cilia in many cell types and organs remains poorly understood.

Cells inherit two centrioles, an older (mother) and a younger (daughter) centriole, that serve as key components for the microtubule organizing center and as foundations for cilia. Cilia assembly is a multi-step process that begins with anchoring of the basal body to the plasma membrane after cells exit the mitotic cycle. These processes are mediated by the distal appendages (DAPs) that project from the distal end of the mother centriole and dock to the cytoplasmic leaflet of the plasma membrane. Their molecular composition, structure, and function in ciliogenesis have been increasingly elucidated and characterized in recent years. To date, six proteins—CEP83, CEP89, SCLT1, CEP164, LRRC45, and FBF1—have been defined by their distal recruitment to the mother centriole to form the core of the distal appendages [15,16,17,18].

Defects in genes’ encoding centrosomal proteins cause an autosomal recessive disorder characterized by dwarfism, microcephaly, and mental retardation (Seckel syndrome) [19,20,21], whereas ciliary dysfunction or their absence contributes to a plethora of human diseases presenting with highly variable clinical manifestations that have collectively been termed “ciliopathies” [22,23]. This complex of disorders can be associated with retinal degeneration, cystic renal disease, obesity, liver dysfunction, skeletal deformities, congenital heart defects, and brain developmental abnormalities [23]. Mutations in genes encoding DAP components have been implicated in the pathogenesis of ciliopathies and reported to exhibit ciliogenesis defects. In this review, we focus on primary cilia and ciliopathies with particular emphasis on DAP proteins and their impact on cellular homeostasis and human diseases.

## 2. Primary Cilia—Basic Structure and Molecular Composition

Ultrastructural studies resolved three subcellular main components of cilia (Figure 1A): the basal body, a specialized centriole that anchors the cilia to the cell body; the axoneme, a microtubule extension comprised of microtubule doublets; and the ciliary membrane, which sheathes the axoneme [16,18].

The DAP proteins have been found to play an important role in the initial steps of ciliogenesis [24,25,26]. Cilia assembly starts when cells exit the cell cycle and become quiescent or differentiate. The transition from the centriole to the basal body is initiated by the accumulation and fusion of Golgi-derived cytoplasmic vesicles to the distal appendages of the mother centriole generating the primary ciliary vesicle. The vesicle then migrates to the cell surface and anchors to the plasma membrane through the distal appendages. After docking to the cell membrane, the transformation into the basal body is completed [27]. The basal body, which consists of nine triplet microtubules, remains connected to the daughter centriole, and functions as the microtubule-organizing center for the cilium [24,25,26].

Distal of the basal body the transition zone is assembled, which acts as a selective barrier making the ciliary compartment biochemically distinct from the rest of the cell. This barrier is established by the transition fibers and by the Y-shaped fibers or ‘Y-linkers’. The transition fibers connect the basal body to the ciliary membrane through the centriolar appendages, whereas the Y-linkers connect the ciliary membrane to the underlying axoneme. Both structures organize the diffusion barrier for membrane-associated soluble proteins regulating their ciliary entry and exit. The transition fibers may also serve as a docking site for intraflagellar transport (IFT) proteins [28].

The ring of nine microtubules triplets of the basal body form a template for the ring of nine microtubule doublets of the ciliary axoneme. The biological basis of converting triplets to doublets between the basal body and the axoneme remains unknown. The microtubules are composed of alpha- and beta-tubulin, which has been post-translationally modified to stabilize it from depolymerization. The axonemal microtubule-based structure confers the function of the cilium as either a motile cilium or non-motile (primary) cilium. In contrast to primary cilia, motile cilia possess an extra central microtubule pair. This “9 + 2” arrangement and the presence of specialized motor proteins enable ciliary motility to create coordinated beating patterns. Motile cilia are present in multiple copies per cell and are found on airway epithelial cells, oviduct cells, and ependymal cells in the brain. Interestingly, nodal cilia also lack the central microtubules, but possess ciliary motor proteins allowing them to generate a rotary motion that is required for body patterning [29].

The ciliary membrane sheathes the axoneme and is connected by the Y-linkers of the transition zone. Although the ciliary membrane is continuous with the plasma membrane it possesses a unique lipid composition and is highly enriched for specific signaling molecules, including transmembrane receptors and signaling phosphoinositides. This structural and physiological composition supports the mechano-sensory ciliary function and its participation in the signal transduction of several molecular pathways.

As cilia lack their own protein synthesis machinery all components need to be transported to the cilia via specialized transport processes. Ciliary assembly and maintenance rely on a polarized trafficking system from the Golgi apparatus and the endocytic recycling compartment to the basal body mediated by small GTPases of the Rab family. Rab8 and its activator Rabin8 are essential for the entry of protein cargoes into the ciliary compartment. Targeting of Rabin8 to the ciliary base is regulated by Rab11. Recently, it has been demonstrated that PtdIns3P selectively produced in the endocytic recycling compartment promotes activation of Rab11a, triggering the translocation of proteins to the primary cilium [30,31]. The bi-directional cargo transport along the axoneme is provided through the intraflagellar transport (IFT) system [32]. The kinesin-2 motor-IFT B complex enables the anterograde shuttle from the basal body to the tip, while dynein-IFT A protein complexes coordinate the retrograde transport [33,34].

The critical role of distal appendages in membrane docking to initiate ciliogenesis is based on various functions. Distal appendages are required to remove the ciliogenesis inhibitor CP110 from the mother centriole by recruiting the Tau tubulin kinase 2 (Ttbk2) [15,18]. Furthermore, the distal appendage protein CEP164 regulates ciliary-directed vesicular transport through its interaction with Rab8 and Rabin8 [35]. At least six proteins are required for the establishment of the distal appendages ensemble: centrosomal protein 83 (CEP83), centrosomal protein 164 (CEP164), centrosomal protein 89 (CEP89), sodium channel and clathrin linker 1 (SCLT1), Fas binding factor 1 (FBF1), and leucine rich repeat containing 45 (LRRC45) [15,18]. The disruption of the DAP complex results in an impaired ciliary assembly and mutations in genes encoding DAP proteins are characterized by phenotypes affecting various organs associated with ciliopathies [36,37,38,39,40,41].

Super-resolution microscopy studies revealed that DAPs are organized in a conical-shaped architecture, which interfaces the centriole and cilium, and the plasma and ciliary membranes. CEP83, CEP89, SCLT1, and CEP164 form the backbone of the radially localized core DAP components, exhibiting a symmetric ring-like pattern [15]. At the root of the windmill-like blades is CEP83, while CEP164 is extended at the tip near the docking site of the membrane (Figure 1B). The gap between adjacent blades is filled with the distal appendage matrix (DAM) containing the core IFT complex component IFT88, the small GTPase ARL13B, and FBF1 [16,18]. Since FBF1 is associated with the proximal junction of the ciliary pocket, a hotspot of ciliary endo- and exocytosis, FBF1 may serve as a barrier to gate the ciliary compartment. The assembly of DAPs occurs in a hierarchical sequence (Figure 1C) [15]. CEP83 and SCLT1 regulate the recruitment of CEP164 and LRRC45 [15,18]. LRRC45 localizes to the distal appendage of the mother centriole as part of the appendage blade complex and aids FBF1 into the matrix between the blades.

More recent data support the presence of a distal centriole complex, which is required to assemble the distal appendages. The complex consists of three proteins, MNR, OFD1, and CEP90. MNR forms the innermost ring at the distal centriole and recruits OFD1, together with CEP90 [42]. CEP90 is a key component of the centriolar satellites and of the distal end of centrioles. CEP90 regulates mother centriole function and recruits CEP83 to initiate distal appendage assembly at the mother centriole [42].

## 3. The Genetic Basis of Nephronophthisis (NPHP) and Related Disorders

The term nephronophthisis-related ciliopathies (NPHP-RC) summarizes a group of autosomal-recessive cystic kidney diseases, including nephronophthisis (NPHP), Senior–Løken syndrome (SLS), Joubert syndrome (JBTS), and Meckel–Gruber syndrome (MKS) [23]. NPHP-RC are genetically heterogeneous disorders, caused by mutations in genes encoding proteins that localize to primary cilia, basal bodies, or centrosomes. Their disruption leads to structurally or functionally aberrant cilia, resulting in a broad phenotypic spectrum, which is collectively termed “ciliopathies” [23,43]. NPHP-RC represent the most frequent monogenic cause of kidney disease in children and young adults that progress to kidney failure within the first three decades of life [44,45]. Commonly, NPHP-RC are accompanied by multiple extra-renal organ manifestations such as retinal degeneration or periportal liver fibrosis [46].

NPHP-RC are considered rare disorders with varying incidences of 0.1–0.2 per 10,000 live births [47,48,49]. However, the overall prevalence of NPHP-RC is likely to be an underestimation. Recent studies indicate that NPHP is a relatively frequent cause of end-stage renal disease (ESRD) in adults [50]. The traditional classification of NPHP is based on the age of onset, distinguishing three forms of progression—infantile, juvenile, and adolescent/adult. The juvenile form is most common and also referred to as classic NPHP. The median age of progression to ESRD is <4 years for infantile NPHP, 13 years for juvenile NPHP, and 19 years for adolescent NPHP [45].

The initial clinical symptoms of NPHP are typically mild and often nonspecific, including polyuria with secondary enuresis and polydipsia due to urinary concentrating defects. In contrast to other kidney diseases affected individuals usually do not develop severe hypertension [51]. The most prominent renal features are increased echogenicity and corticomedullary cysts with normal or small-sized kidneys on renal ultrasound [44,52]. Hallmarks of NPHP in renal histology are thickening and disintegration of the tubular basement membrane, interstitial fibrosis and tubular atrophy, and cyst formation [53], as shown in Figure 2. Corticomedullary cysts are recorded in 70% of patients with juvenile NPHP [54]. Electron microscopy may reveal tubular basement membrane duplication and thickening. It should be noted that another rare kidney disease, autosomal dominant tubulointerstitial kidney disease (ADTKD), shares similar histological features. Therefore, the histopathological diagnosis is superseded by the genetic diagnosis.

About 15% of the affected individuals with NPHP-RC exhibit extrarenal manifestations. Its recognition is important and may allow defining a clinical diagnosis. The most notable extrarenal manifestation, retinal degeneration, occurs in 10% of all patients with NPHP and is defined as Senior–Løken syndrome (SLS) [55,56,57]. Cerebellar malformations such as cerebellar vermis hypoplasia/aplasia are commonly observed in individuals with Joubert syndrome (JBTS) [58,59], characterized by the “molar tooth sign” in brain imaging studies [60]. These malformations are associated with developmental delay, intellectual disability, muscle hypotonia, ataxia, and oculomotor apraxia. Meckel–Gruber syndrome (MKS) represents the most severe manifestation due to extreme multiorgan involvement that often results in perinatal lethality [61]. Bardet–Biedl syndrome (BBS) is a complex, genetically heterogeneous disorder that affects numerous organ systems. Mutations in 19 different genes have been described as causative, leading to intellectual disability, obesity, cystic kidney disease, retinitis pigmentosa, and polydactyly [62,63].

To date, up to 90 genes have been implicated in the pathogenesis of NPHP-RC, and mutations in 25 of these genes cause NPHP [49]. Many of the gene products participate in functional protein complexes, which localize to primary cilia or to centrosomes. Pathogenic variants in these genes lead to variable defects [56], resulting in clinical symptoms such as retinal, kidney, and neurodevelopmental defects, as well as early lethality, summarized as ciliopathies [46].

Mutations in *NPHP1* are the most common cause of NPHP (MIM 256100), accounting for 20–25% of all cases. About 85% of these cases are based on homozygous full gene deletions of the NPHP1 gene, which have also been identified in affected individuals with SLS (MIM 266900), and very rarely in those with JBTS (MIM 609583). Interestingly, a recognizable genotype–phenotype relationship has not been detected. The gene product, nephrocystin-1 localizes to adherens junctions, focal adhesions, and to the ciliary transition zone. Although deletions in NPHP1 have already been described in 1997, the pathophysiological mechanisms by which NPHP1 disruption leads to NPHP remain unknown. Moreover, the physiological function of Nephrocystin-1 is uncertain. However, high-confidence proteomic [64] and proximity-dependent biotinylation studies have shown that Nephrocystin-1 interacts directly with other NPHP proteins [65]. Thereby, the ciliary proteins form an interaction network, which can be classified into three biochemically distinct modules: (1) the NPHP1-4-8 module; (2) the NPHP5-6 module; and (3) the MKS complex.

The NPHP1-4-8 module, consisting of NPHP1, NPHP4, and NPHP8, localizes to the ciliary transition zone and may play a role in epithelial morphogenesis and in the establishment of tissue architecture. Mutations in *NPHP4* have been identified in affected individuals diagnosed with juvenile NPHP (MIM 606966) and SLS (MIM 606996). Interestingly, NPHP4 has been shown to regulate Hippo signaling through promoting phosphorylation and nuclear shuttling of YAP/TAZ [66]. Furthermore, NPHP1 directly interacts with NPHP3, which localizes to the Inversin compartment (IC) within the proximal cilium [67]. The IC consists of at least three additional proteins: Inversin/NPHP2 (INVS), Serine/threonine-protein kinase NEK8/NPHP9, and ANKS6/NPHP16 [68]. Pathogenic gene variants encoding proteins of the IC cause a spectrum of overlapping phenotypes in humans and rodent models, including cystic kidney disease, cardiovascular abnormalities, and periportal liver fibrosis [68,69,70,71,72,73,74]. Although its assembly hierarchy is characterized, and a link to Wnt and Hippo-signaling established, the role of IC-mediated signaling events remains incompletely understood [68,69,70,71,72,73]. NPHP8/RPGRIP1L controls the assembly of the ciliary transition zone and regulates the ciliary gating function [75]. Consequently, its deficiency leads to an altered ciliary protein composition [76]. Moreover, RPGRIP1L regulates the proteasomal activity at the primary cilium by interacting with Psmd2, a component of the regulatory proteasomal 19S subunit [77]. In addition, RPGRIP1L has been shown to be essential for Hedgehog signaling responsiveness and to regulate mTOR-mediated autophagic activity [78,79]. Mutations in *NPHP8* lead to severe ciliopathies, including MKS (MIM 611561) and JBTS (MIM 611560) [80].

The NPHP5-6 module encompasses NPHP5 and NPHP6 and localizes to the centrosome. Mutations in *NPHP5/IQCB1* cause a retinal–renal phenotype characterized by retinitis pigmentosa with NPHP [55]. The binding partner, NPHP6/CEP290, is one of the most intriguing NPHP-RC-associated disease genes. So far, more than 100 different mutations in *CEP290* have been identified, causing a broad variety of distinct phenotypes, including SLS (MIM 610189), JBTS (MIM 610188), and MKS (611134) [81]. The subcellular localization of CEP290 is regulated in a cell cycle-dependent manner. In quiescent cells, CEP290 is an integral component of the ciliary transition zone, whereas is in dividing cells it localizes to the distal mother centriole [82]. A potential role for NPHP5/NPHP6 in primary cilium assembly was established by identifying CEP290 as essential for targeting Rab8a, which is required for ciliary growth through the initiation of intraciliary and vesicular trafficking [82].

The MKS module includes MKS1 and its interacting proteins, which localize to the base of the cilium. This complex is characterized by its connection to Hedgehog signaling-mediated neural tube development and binds Tectonic2 (TCTN2) [67,83]. Mutations in genes encoding for proteins of this complex lead to MKS (MIM 249000) [84], a severe pleiotropic autosomal recessive developmental disorder characterized by developmental defects of the central nervous system that include neural tube defects.

Although nephronophthisis-related ciliopathies (NPHP-RC) are among the most frequent monogenic causes of kidney disease during the first three decades of life, therapeutic options are almost nonexistent. As a result, patients are at a greatly heightened risk of kidney failure and requirement of renal replacement therapy. Despite our improved understanding about the relevance of cilia, an ever-increasing number of established ciliopathy-associated genes and improved genetic diagnostics, the pathomechanisms underlying NPHP-RC remain incompletely characterized. This highlights the urgent need to explore the underlying disease mechanisms and to identify new therapeutic targets.

## 4. Molecular Functions of DAPs in Ciliogenesis

Cilia are observed primarily in quiescent or differentiated cells in both developing and adult tissues. Ciliogenesis is coupled to the cell cycle and occurs from the distal end of the mother centriole as cells exit the mitotic cycle at the G1/G0 phase [85]. Ciliary assembly is an elaborately regulated process involving various cellular machineries and signaling pathways. Ciliogenesis starts with the docking of the basal body to the plasma membrane, tightly mediated by the distal appendages (DAPs) of the mother centriole that involves the orchestrated recruitment of six DAP proteins [86]. In addition to anchoring the mother centriole to the cellular membrane, DAPs also play essential roles in recruitment of TTBK2, in Rab8a-targeted vesicle trafficking, and in centriolar satellite organization [18] (Figure 3). Furthermore, DAPs create a border between the plasma membrane and the ciliary membrane, thus functioning—together with nucleoporins and the ciliary transition zone—as a gate enabling the selective translocation of proteins into the ciliary compartment.

Cilia formation starts when Golgi- and recycling endosome-derived vesicles, termed distal appendage vesicles, accumulate in proximity to the distal appendages and dock to the mother centriole [87]. These vesicles then fuse, forming the primary ciliary vesicle (PCV), mediated by EHD1 and SNAP29 [88,89]. Following assembly of the PCV, Tau Tubulin Kinase 2 (TTBK2) is recruited, which allows to remodel the distal ends of the basal bodies by removing the CP110–CEP97 complex. CP110 ‘caps’ the basal body and functions as a negative regulator of ciliary extension and as a checkpoint against inappropriate cilia formation. Vesicular transport to the centrosome includes the recruitment of the Rab11/Rab8/Rabin8 complex, which promotes cilia membrane biogenesis. Rab8 and its activator, Rabin8, are fundamental for the transport and entry of proteins into the ciliary compartment. Targeting of Rabin8 to the ciliary base is regulated by the Rab GTPase Rab11, a coordinator of endosome recycling to the plasma membrane. At the ciliary base, Rab11 directly associates with Rabin8 mediating the guanine nucleotide exchange of Rab8 [90,91]. Additionally, the IFT machinery coordinates the bidirectional transport of the ciliary proteins along axonemal microtubules during cilia assembly and maintenance [92]. DAPs also play a critical role in centriolar satellite organization, which is essential for TZ establishment and positioning of Rab8 vesicles at the mother centriole upon cilia formation [93,94,95] (Figure 3).

Dysfunction of the distal appendages causes diverse developmental disorders in humans and animal models. Since DAPs are key components of microtubule-organizing center and provide the foundation for ciliary assembly, deficient or defective DAP proteins are associated with disrupted ciliogenesis and alterations in the cilia-related signaling pathways. To date, human mutations have been reported in three genes encoding DAPs proteins (CEP83, CEP164, and SCLT1) and in CEP90, which recruits the proximal-distal appendage component CEP83. Various genetic studies of model systems such as mutant mouse models emphasize that DAP deficiency is associated with the pathogenesis of ciliopathies. In the following sections, we will review the detailed function of the six DAP proteins for cilia assembly and their implication in disease development of NPHP-RC.

### 4.1. CEP83/CCDC41/NPHP18

CEP83 is the most proximal DAP protein, recruited by CEP90 to initiate distal appendage formation [16]. Loss of CEP83 in retinal pigment epithelial cells (RPE-1) and murine inner medullary collecting duct (IMCD3) cells prevents the recruitment of all other DAP components but does not affect their expression levels. In cells lacking CEP83, the primary ciliary vesicle fails to anchor to the mother centriole, leading to defective initiation of ciliogenesis [15,96,97]. This indicates that CEP83 is essential to initiate cilia formation. Interestingly, studies on human telomerase-immortalized retinal pigment epithelial cells (hTERT-RPE1) have demonstrated that TTBK2-mediated phosphorylation of CEP83 regulates ciliary vesicle docking and CP110 removal. CP110 removal also depends on the degradation of M-Phase Phosphoprotein 9 (MPP9), a substrate of TTBK2 [98,99]. TTBK2-dependent MPP9-phosphorylation enhances its degradation, which promotes cilia initiation. It has further been shown that MPP9 degradation and subsequent CP110 removal depend on the recruitment of a cilia-specific E3 ligase to the mother centriole [99]. This process is accompanied by TTBK2-dependent CEP83 phosphorylation and changing of CEP83 conformation (Figure 4A) [97]. MPP9 is recruited to the distal end of the mother centriole by the Kinesin Family Member 24 (KIF24), enhancing the recruitment of CP110–CEP97 by binding to CEP97. Morpholino-mediated knockdown of the CEP83 ortholog Ccdc41 in zebrafish leads to olfactory ciliogenesis defects. The removal of CEP83 from radial glial progenitor cells in mice disrupts the anchorage of the centrosome abolishing cilia formation and leads to an excessive proliferation with an enlarged cortex formation, and activation of the Hippo signaling key effector protein YAP [96].

In humans, recessive mutations in *CEP83* (OMIM 615847) were identified as the molecular cause for Nephronophthisis-18 (NPHP18; MIM 615862) [36]. To date, nine patients from eight independent families with homozygous or compound heterozygous mutations in the *CEP83* gene have been reported. Five affected individuals carried compound heterozygous mutations composed of a missense mutation and either an in-frame deletion or a protein truncating mutation. Three families with homozygous mutations have been identified: One with a missense, one with an in-frame deletion, and one carrying a truncating mutation. All affected individuals showed an early-onset nephronophthisis resulting in end-stage renal disease at 1 to 4 years of age. Different histological alterations of the kidney were described in individuals with *CEP83* mutations [36]. Three individuals displayed microcystic tubular dilatations, one individual had glomerular cysts and glomeruli dysplasia, and two individuals had abnormal thickness of the tubular basement membranes. Interstitial fibrosis was observed in five patients. Extra-renal manifestations, including neurological alterations, such as intellectual disability, and/or hydrocephalus, have been detected in four individuals with *CEP83* mutations [36], as referred in Table 1. Two individuals presented with periportal liver fibrosis. The most severe phenotype has been observed in one affected individual with a homozygous truncating mutation of *CEP83* accompanied by triple X syndrome and included ESRD, facial dysmorphism, and heart anomalies [36]. Patient-derived fibroblasts from two individuals carrying one truncating mutation in trans with either a missense or an in-frame variant showed a decreased percentage of ciliated cells and an altered subcellular distribution of CEP164, while the localization of CEP89 remained unaffected. *CEP83* mutants that represented mutations, leading to a truncated protein or to an in-frame deletion of amino acids in the coiled-coil domains of CEP83, failed to localize to the centrosome and accumulated in the nuclei when transfected into RPE1 cells. Furthermore, these *CEP83* mutants failed to interact with CEP164 and *IFT20*. In contrast, missense variants of CEP83 and in-frame deletions outside the coiled-coil domains did not display defects of centrosomal localization.

### 4.2. CEP164/NPHP15

CEP164 localizes in a cell cycle-dependent manner to the mother centriole and to mitotic spindle poles [16]. Super-resolution studies revealed that CEP164 comprises part of the backbone structure of the DAP blades [16]. Its recruitment to the centriole depends on CEP83 and SCLT1. Depletion of CEP164 in mammalian cells leads to the reduced anchorage of the mother centriole to the membrane and failed targeting of Rab8a to the centrosome abolishing cilia formation [15,109,110]. This implies a model where CEP164 binds to the Rab8a/Rabin complex, promotes the accumulation of Rab8 at the centrosome, and regulates the docking of Golgi-derived vesicles to the distal appendages [110,111]. Indeed, the proper centriolar migration and docking depends on the formation of a CEP164/Rabin8 complex, enhanced by the coiled protein Chibby (Cby). After binding to CEP164, Cby recruits Rabin8, thereby promoting the formation of the CEP164/Rabin8 complex that leads to polarized transport and fusion of Rab8-positive vesicles to ensure maturation of ciliary vesicles, which enables the anchorage of the basal body to the apical membrane (Figure 4B) [111]. Moreover, CEP164 recruits TTBK2, which triggers ciliogenesis through targeting IFT components and releasing CP110 from the mother centriole [112].

CEP164 deficiency has also been associated with an impaired DNA damage response (DDR) and altered cell cycle checkpoint control. In hTERT-RPE cells, CEP164 localizes to the nucleus in proximity with DDR proteins, such as SC-35 (a splicing factor), checkpoint kinase 1 (CHK1), and Tat-interactive protein 60 (TIP60) [102], and interacts with the cell cycle checkpoint proteins ataxia telangiectasia mutated (ATM) and ataxia telangiectasia and Rad3-related protein (ATR) during activated DDR signaling. Accordingly, depletion of cep164 in zebrafish causes sensitivity to DNA damage as evidenced by an increased expression of phosphorylated γH2AX [102]. Moreover, CEP164 deficiency leads to a delayed S-phase progression and an abrogation of the G2/M checkpoint, suggesting an essential role in cell cycle regulation. IMCD3 cells depleted of Cep164 show evidence of apoptosis and epithelial-to-mesenchymal transition (EMT) [39].

Human recessive mutations in *CEP164* (OMIM 614848) lead to Nephronophthisis-15 (NPHP15; MIM 614845) [102,103] and presumably to Bardet–Biedl syndrome [104] and primary ciliary dyskinesia (PCD) [105]. Four affected individuals have been reported with homozygous truncating mutations in *CEP164*, and one with a homozygous missense mutation. Furthermore, two individuals displayed compound heterozygous mutations consisting of a truncating mutation in trans with a missense variant or two missense mutations. While hypomorphic mutations lead to NPHP and Senior–Løken syndrome, null mutations have been reported to cause a more severe dysplastic phenotype of Meckel syndrome and Joubert syndrome, as shown in Table 1. The overexpression of *CEP164* constructs, which mimic two of the detected truncating mutations in IMCD3 cells, abrogated the subcellular localization of CEP164 at the mother centriole. In contrast, the transfection with constructs reflecting two of the missense mutations showed a proper centrosomal localization of CEP164 in hTERT RPE-1 cells but lead to diminished cilia formation. Interestingly, these constructs also compromised the interaction with TTBK2 through an impaired folding of the N-terminal domain of CEP164 [113]. The impaired S-phase progression upon depletion of CEP164 in IMCD3 cells cannot be rescued through the overexpression of two different disease-associated cDNA-constructs that mimics a nonsense and a missense human mutation. Furthermore, the transfection of the same nonsense *CEP164* allele induced epithelial-to-mesenchymal transition and an increased expression of pro-fibrotic genes [39]. The knockdown of *cep164* in zebrafish embryos mimics ciliary phenotypes, including laterality defects, pronephric tubule cysts, and retinal dysplasia [102]. Global Cep164 deficiency in mice leads to early embryonic lethality due to holoprosencephaly, cardiac looping defects, and a truncated posterior trunk [106]. A collecting duct-specific deletion of Cep164 abolishes ciliogenesis and leads to a dysregulated cell cycle and epithelia cell hyperproliferation that drives renal cyst growth [40]. Interestingly, treatment of these mutant mice with a cyclin-dependent kinase inhibitor reduces cortical cyst formation and epithelial proliferation, restoring normal cortical histology [40]. Depletion of Cep164 in multiciliated cells of FOXJ1-positive tissues results in hydrocephalus and perturbs the differentiation of multiciliated cells in murine airway and oviduct epithelia [106].

It remains unclear whether the ciliary or nuclear dysfunction predominantly accounts for the renal and extra-renal disease development in individuals with deficient CEP164. Considering the phenotypic overlap with affected individuals carrying mutations in other NPHP-RC-associated genes whose products localize in different ciliary compartments lacking a nuclear localization, it is tempting to speculate that the ciliary dysfunction primarily determines the underlying pathogenesis [26]. However, it should also be noted that many cilia proteins have extra-ciliary functions, such as cell cycle regulation [114,115]. Therefore, one can hypothesize that, independently of the ciliary dysfunction and presumably dependent on genotype, the nuclear dysfunction of CEP164 promotes disease progression through an altered cell cycle regulation and DDR response, thus explaining the phenotypically heterogeneity [102]. Further functional studies are needed to address these hypotheses.

### 4.3. CEP89/CCDC123/CEP123

CEP89 localizes to the distal end of the centriole within the appendage blade complex recruited through CEP83, independently of SCLT1 [16]. Depletion of CEP89 in RPE-1 cells results in disrupted ciliary vesicle formation and failed docking to the distal mother centriole, leading to defective ciliogenesis [15,116]. CEP89 interacts with the centriolar satellite proteins PCM-1 (pericentriolar material 1), OFD1 (oral-facial-digital syndrome 1), BBS4 (Bardet–Biedl Syndrome 4), and CEP290 (Centrosomal protein 290) [116] (Figure 4C). *PCM-1* and CEP290, in turn, interact physically and functionally, recruiting the small GTPase Rab8a, a key regulator of vesicle trafficking. Rab8a mediates the membrane-trafficking to centrosomes and cilia, promotes anchorage and fusion of membranous vesicles for cilia assembly, and is required for ciliary elongation and maintenance [117,118]. A more recent systematic mapping of the centrosome–cilium interface confirmed this protein network and showed that the poorly studied peptide C3orf14 also interacts directly with CEP89 [65]. Further studies on mouse 3T3 cells have demonstrated that the interacting partners CEP290 and Rab8a form a distinct complex complemented by CP110, which antagonizes the activity of CEP290 and prevents CEP290-dependent Rab8a ciliogenesis [119]. It is therefore tempting to speculate that CEP89 may also play a critical role in migration and insertion of mother centrioles into the plasma membrane, implementing cilia assembly. Further evidence for this participation comes from the association with PCM-1, a central scaffold for the binding of centriolar satellite proteins. PCM-1 depletion leads to impaired primary cilia formation. In addition, PCM-1 organization is impaired in cells lacking CEP89. Interestingly, deficiency in OFD1 and BBS4 also results in diminished ciliogenesis [120] through disrupted distal appendage formation, impaired centriole elongation [121], and PCM-1 mislocalization [122].

### 4.4. LRRC45

LRRC45 is also part of the appendage blade complex, localizes to the proximal end of both the mother and daughter centrioles, and is associated with the basal body of primary and motile cilia [18,123]. LRRC45 is recruited to the centriole by CEP83 and SCLT1. LRRC45 is needed to shuttle FBF1 into the distal appendage matrix [18]. Consequently, the depletion of LRRC45 leads to decreased levels of FBF1 at the mother centrioles [18]. RPE-1 cells lacking LRRC45 show an impaired cilia formation, indicating that LRRC45 is needed for cilia assembly and elongation [18]. Further studies have demonstrated that LRRC45 is necessary for cilia-directed trafficking and docking of Rab8-postive vesicles—similar to the presumed function of CEP89, but not for the Cep164/TTBK2-dependent removal of the CP110–Cep97 complex [18]. LRRC45 also plays a critical role in organizing centriolar satellite proteins, including PCM-1 and SSX2IP, the latter promoting centrosome maturation [18]. Additional studies on HeLa and U2OS cells indicate that LRRC45 functions as a centrosome linker, and its depletion leads to centrosome splitting during the interphase [123].

### 4.5. FBF1

FBF1 is recruited to the centriole by SCLT1. FBF1 localizes within the distal appendage matrix between the appendage blades in close proximity to the ciliary membrane [15,16,17]. FBF1 depletion in RPE-1 cells results neither in an impaired CP110-TTBK2 recruitment nor in defective ciliogenesis [18]. It is hypothesized that FBF1 plays a critical role in ciliary gating and arranging the ciliary composition. This is supported by the reduced distribution of IFT88 in the DAM, leading to a mislocalization of ciliary transmembrane proteins. In agreement, studies on the *C. elegans* homolog dyf-19 have demonstrated a regulatory function of FBF1 for the entry and transit of IFT particles into the cilium [124,125] (Figure 4D). Similar to LRRC45, the depletion of FBF1 results in a satellite disorganization, as evidenced by a decreased concentration of the proteins PCM1 and SSX2IP [18].

### 4.6. SCLT1

SCLT1 belongs to the backbone of the pinwheel-like DAP blades and is required for the recruitment of CEP164, FBF1, and LRRC45 to centrioles [15,16,17,18]. Depletion of SCLT1 in cultured RPE1 cells results in a lack of cilia formation [15]. Likewise, loss of Sclt1 in mice leads to disrupted cilia assembly, causing polydactyly and renal cyst formation, including an increased proliferation and apoptosis rate in murine tubular epithelial cells [38]. Further studies have demonstrated that Sclt1 deficiency promotes ERK signaling through the PC2/cAMP/PKA axis and activates the STAT3 and TGF-β/SMAD pathways in renal epithelial cells [38]. Consistently, the cystic kidney phenotype could be partially rescued by the administration of a STAT3 inhibitor [38]. Interestingly, similarly to Cep164 loss of function, cyst growth in the kidney has been attributed to tubular hyperproliferation along with increased expression levels of cell cycle markers. Furthermore, and equivalent to the other members of the DAP complexes, SCLT1 plays a critical role in centriolar satellite organization, since SCLT1 depletion in RPE-1 cells alters the expression levels of the centriolar satellites proteins PCM-1 and SSX2IP [18].

In humans, compound heterozygous mutations in *SCLT1* (OMIM 611399) have been detected in an affected individual diagnosed with Senior–Løken syndrome. Both variants located at the exon–intron boundaries and have demonstrated to generate an aberrant splicing transcript [37]. In addition, one homozygous splice site mutation causing a protein truncating product has been reported in one individual diagnosed with orofaciodigital syndrome (OFD IX, MIM 258865) who presented with midline cleft, microcephaly, choanal atresia, and coloboma, as well as congenital heart involvement [107]. Compound heterozygous pathogenic missense variants were detected in two independent individuals with Bardet–Biedl syndrome, who presented with renal dysfunction, intellectual disability, short stature, and truncal obesity [108], as summarized in Table 1. It remains unclear to what extent the reported mutations affect the function and localization of the *SCLT1* gene products. Based on the genetic findings, a loss of function can be assumed for the affected individuals with SLS and OFD IX.

### 4.7. CEP90/PIBF1

CEP90 has been identified as a component of centriolar satellites and previously as part of the distal centriole complex (DISCO) [126], which serves as a foundation for DAP formation. CEP90 is essential for spindle pole integrity, and for the recruitment of *CEP63* as well as of CDK2, a cyclin-dependent kinase with well-known functions for cell cycle progression and centriole duplication [127]. Furthermore, CEP90 interacts with centriolar satellite protein PCM1, which delivers proteins to the centrosome and with MNR and OFD1 [128]. Together with the latter, CEP90 forms DISCO, which recruits CEP83 to root distal appendage formation. Consequently, depletion of CEP90 leads to mitotic arrest, misaligned chromosomes, spindle pole fragmentation, and centriole duplication defects [129,130]. Moreover, cells lacking CEP90 fail to remove CP110 and to recruit the DAP proteins CEP83, FBF1, SCLT1, and CEP164 [42].

In humans, biallelic mutations in *CEP90*, also known as *PIBF* (OMIM 607532), have been reported in seven independent families diagnosed with Joubert syndrome (MIM 617767) [131,132]. The affected individuals presented with cystic kidney disease, liver fibrosis, retinal dystrophy, and distinctive brain malformation. Morpholino oligomer-mediated knockdown of the orthologous gene in a Xenopus model resulted in ciliogenesis defects and impaired motility of multiciliated cells [131]. Mice carrying a homozygous deletion of seven exons in *Cep90* display cardiac looping defects, leading to early embryonic lethality. Further studies indicate that embryos and MEFs lacking Cep90 fail to assemble distal appendage-like structures, resulting in defective cilia assembly and disrupted Hedgehog signaling [42]. Interestingly, mutations in genes encoding for the other two components of the distal centriole complex, MNR/KIAA0753 (OMIM 617112), and OFD1 cause JBTS (MIM 300804) or orofaciodigital syndrome (MIM 311200) [42].

## 5. Conclusions

DAPs are involved in multiple essential biological processes, including ciliogenesis, cell cycle regulation, and DNA damage response. In addition, DAPs regulate cellular responses such as cell proliferation, epithelial-to-mesenchymal transition, and apoptosis [15,18,97,109,110,111,116]. The present cell-based studies impressively demonstrated that the hierarchically organized DAPs play a crucial role in the early phase of cilia formation. They influence and regulate both the earliest initiative steps of primary vesicle formation, the migration and anchoring of the centriole to the basement membrane, and the recruitment and modification of the coordinating transport components, outside and inside the cilium. In addition, there is a strong indication that the members of the DAP complex significantly influence the cell cycle regulation and thus also coordinate and regulate essential extra-ciliary biological function.

Mutations in three genes encoding DAP proteins, namely, *CEP83*, *CEP164*, and *SCLT1*, have been reported to be linked to heritable ciliopathies, such as NPHP and Joubert syndromes. Genetic animal models of DAPs have been associated with defective cilia formation and renal cyst development, but with an increased proliferation of renal epithelia cell and dysregulated cell cycle activity. Although previous studies in various mammalian cells and genetically modified animal models have highlighted the importance of DAPs in the pathogenesis of NPHP-RC, their underlying pathomechanisms are largely unknown. To what extent deficient DAP proteins affect ciliary-associated signal transduction pathways, cellular or vesicular transport processes, or specific signaling molecules, including signaling phosphoinositides, remains elusive.

Despite our improved understanding of the relevance of cilia, an ever-increasing level of established ciliopathy-associated genes, and genetic diagnostics, the pathomechanisms underlying NPHP-RC remain unclear, which has hindered the development of therapeutic options. Accordingly, a specific treatment is virtually nonexistent. An improved understanding of the molecular consequences of perturbation in DAP functions may contribute to the development of new treatment approaches.

## Figures and Tables

**Figure 1 ijms-22-12253-f001:**
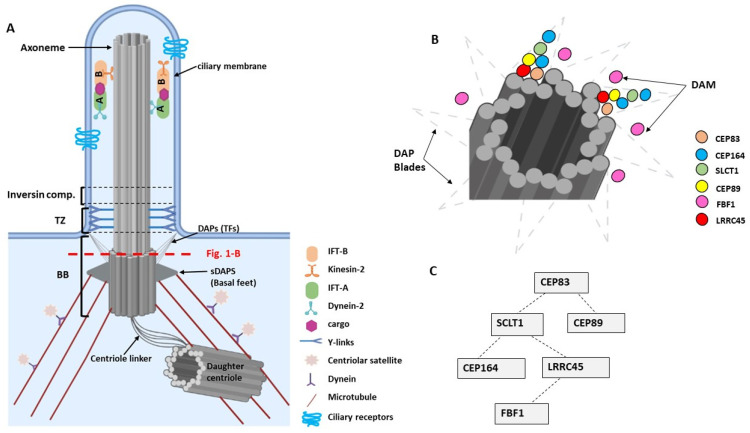
(**A**). Basic structures of a primary cilium. The base of the cilium is composed of a basal body (BB), a transition zone (TZ), and the inversin compartment. The BB consists of a modified mother centriole that is linked to microtubules via subdistal appendages (sDAPs) and that is tethered to the plasma membrane via distal appendages (DAPs) and transition fibers (TFs). The non-continuous red line indicates the section of the centriole at the level of the DAPs that is shown in Figure 1B. (**B**). Schematic diagram showing the arrangement of DAPs as DAP blades comprise CEP83, CEP164, SLCT1, CEP89, and LRRC45 proteins, and the DAP matrix (DAM) includes the FBF1 protein [16]. (**C**). Hierarchy of assembly of the DAPs at the base of the centriole (adapted from [15,18]). Image was created by BioRender.

**Figure 2 ijms-22-12253-f002:**
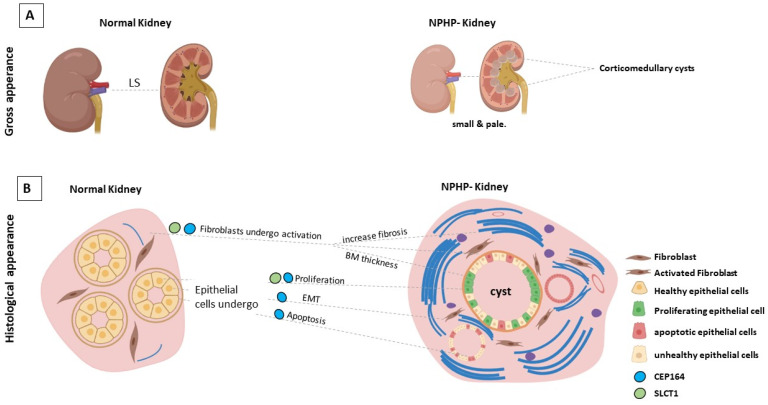
(**A**). Longitudinal section of a diseased NPHP kidney compared with a section of a normal kidney. Cysts form at the corticomedullary junction. (**B**). Diagram depicting a cross section of the kidney, illustrating the multiple cellular stress changes associated with a SCLT1 and CEP164 deficiency during NPHP development. Image was created by BioRender.

**Figure 3 ijms-22-12253-f003:**
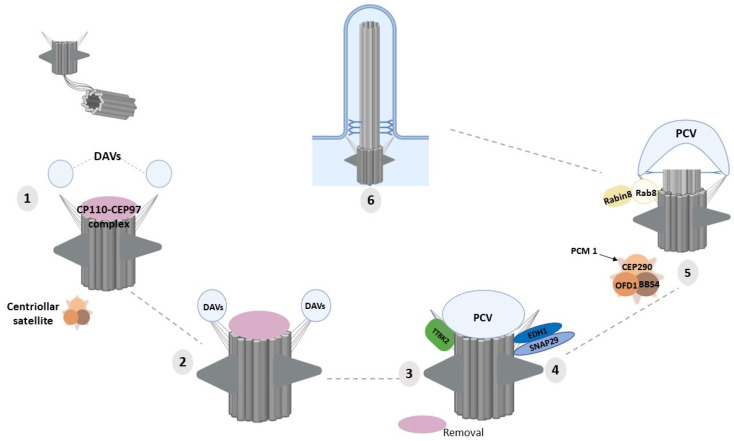
The main steps of primary ciliary vesicle docking to DAPs during ciliogenesis. (**1**) After the conversion of the mother centriole into the basal body, the CP110–CEP97 protein complex caps the basal body and hinders ciliogenesis initiation. (**2**) Docking of small ciliary vesicles to DAPs leads to the formation of distal appendage vesicles (DAVs). (**3**) Larger primary ciliary vesicles (PCVs) form from smaller DAVs (facilitated by EDH1 and SNAP29 proteins). (**4**) Recruitment of TTBK2 and removal of the inhibitory CP110–CEP97 complex. (**5**) Rab8-positive vesicle docks at the centrosome, membranes are extended, and the axoneme is elongated, leading to (**6**) the completion of cilium formation. Image was created by BioRender.

**Figure 4 ijms-22-12253-f004:**
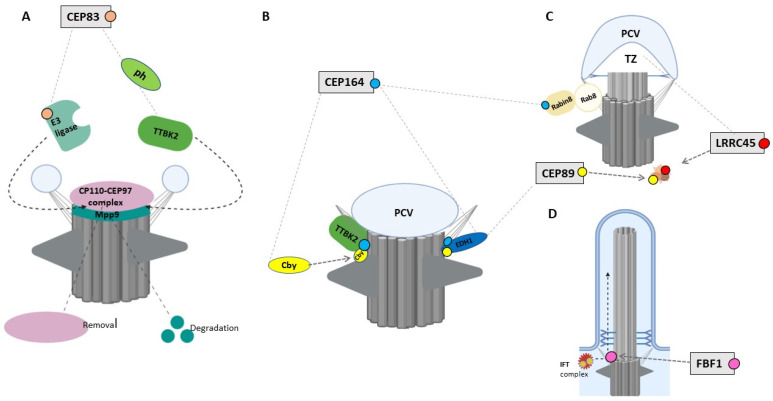
The role of DAPs in ciliogenesis. (**A**). CEP83 recruits E3 ligase and phosphorylates TTBK2 to remove the CP110–CEP97 complex and induce MPP9 degradation. (**B**). CEP164 has three roles: (1) the formation of the CEP164–Cby complex to recruit TTBK2; (2) the recruitment of EDH1 to enhance PCV formation; and (3) the interaction with Rabin8 to activate Rab8. (**C**). LRRC45 and CEP89 organize centriolar satellites. CEP89 plays a direct role in EDH1 recruitment. (**D**) FBF1 regulates the lateral diffusion of the IFT complex. Image was created by BioRender.

**Table 1 ijms-22-12253-t001:** Summary of the current knowledge of DAP genes in relation to human molecular genetics, including its phenotypic description and available animal models.

Gene	Mutations	Renal Phenotype	Extra-Renal Manifestations	Cilia Phenotype	Ref.
* **CEP83** *	*Hs* In 9 individuals diagnosed with NPHP-RC homozygous or compound heterozygous *CEP83* mutations has been identified. 1 individual has been reported with a homozygous missense, 1 with a homozygous protein-truncating mutation. 7 individuals carry at least one loss of function allele.	Nephronophthisis, Tubulointerstitial nephritis, Corticomedullary cysts, Tubular atrophy, and End-stage renal disease.	Eye (in some patients): Retinitis Strabismus Liver (in some patients): Cholestasis, Hepatic cytolysis, Portal fibrosis Central Nervous System (in some patients): Intellectual disability, Hydrocephalus.	(a) primary fibroblasts:impaired ciliation (b) renal biopsy sample: increased ciliary length (c) overexpression of disease construct in RPE-1/ IMCD3: abolished centrosomal localization of CEP83 abrogated protein interaction with CEP164 and IFT20 nuclear accumulation of CEP83 (d) depletion in RPE-1: abolished cilia formation.	[36]
** *Cep83* **	*Mm* Selective deletion of Cep83 in cortical radial glial progenitors (RGPs).		Enlarged brain with abnormal folding.	RGPs lacking Cep83 display a lack of primary cilia.	[96]
* **cep83** *	*Dr* Morpholino-mediated knockdown of ccdc41 (CEP83 ortholog in Zebrafish).		No defect in left/right body asymmetry was observed.	Olfactory placodes showed reduction in cilium formation.	[100]
** *CEP164* **	*Hs* Homozygous and compound heterozygous has been identified in 4 individuals with NPHP-RC. 2 individuals carried loss of function mutations. Compound heterozygous missense mutations lead to Bardet–Biedl syndrome in 1 individual and a homozygous loss mutation to primary ciliary dyskinesia (PCD) in 1 individual.	Nephronophthisis	Eyes: Retinal degeneration, Leber congenital amaurosis, Nystagmus (in 2 patients) Liver (in some patients): Liver failure Central Nervous System: Developmental delay (in 1 patient), Seizures (in 1 patient), Cerebellar vermis hypoplasia (in 1 patient) Skeletal: Polydactyly (in 2 patients) Obesity (in 2 patients) Short stature (in 1 patient) Bilateral bronchiectasis (in 1 patient).	(a) Overexpression of disease construct in IMCD3 cells: abolished centrosomal localization (b) Overexpression of disease construct in hTERT RPE: compromise interaction with TTBK2 (c) Depletion in RPE-1: abolish cilia formation.	[101,102,103,104,105]
** *cep164* **	*Dr*	Pronephric tubule cysts.	Abnormal heart looping, hydrocephalus, and retinal dysplasia.	n/a	[102]
* **Cep164** *	*Mm*	Only in the collecting duct-specific deletion of *CEP*164 mice: Cystic kidneys [40].	(a) Global *CEP*164 deficiency mice: early embryonic lethality, holoprosencephaly, cardiac looping defects, and a truncated posterior trunk [106]. (b) collecting duct-specific deletion of CEP164 mice: only renal cyst growth. (c) CEP164 loss in FOXJ1-positive tissues in mice: results in hydrocephalus.	(a) Global deficiency: abolish cilia formation in neuronal tube (b) Collecting duct-specific deletion: abolishes primary cilia formation in epithelial cells (c) FOXJ1-specific deletion:reduction number of l multiciliated cells.	[40,106]
** *SCLT1* **	*Hs* Compound heterozygous missense mutations has been reported in 1 individual with oro-facio-digital syndrome. Biallelic loss of function mutations have been identified in 1 individual withSenior-Løken syndrom, and in 1 individual with Bardet–Biedl syndrome.	Nephronophthisis (1 patient) bilateral hyperechogenicity, cortico-medullary renal cysts (1 Patient) ESRD at 11 years of age (1 Patient).	(a) Orofaciodigital syndrome type IX (OFD type IX) (2 patients): midline cleft, microcephaly, colobomatous microphthalmia/ anophthalmia, polydactyly, absent pituitary, and congenital heart disease. (b) Senior–Løken syndrome (1 patient): Nystagmus, hepatic dysfunction, megacystis, mild learning disability, autism, obesity (c) Bardet–Biedl syndrome (2 patients): intellectual disability, autism, and motor developmental delay, hepatic fibrosis, short stature, truncal obesity, retinitis pigmentosa.	(a) Depletion in RPE-1: abolish cilia formation.	[107,108]
**Sclt1**	*Mm*	Cystic kidneys	Cleft palate and polydactyly.	Global deficiency: disrupted cilia assembly.	[37,38,108]

Note: Hs, Homo sapiens; Mm, Mus musculus; Dr, Danio rerio.

## Data Availability

The data that support the findings of this study are available from the corresponding author upon reasonable request.

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
