# Peer review of "The Role of Centrosome Distal Appendage Proteins (DAPs) in Nephronophthisis and Ciliogenesis"

_ijms, 2021, doi:10.3390/ijms222212253_

Round 1
Reviewer 1 Report
This is a well-structured, comprehensive review of distal appendage proteins in nephronophthisis. This review not only summarizes recent advances but also will motivate further investigation of many significant unanswered questions about the mechanism and the role of DAPs in ciliogenesis. I found the current manuscript highly significant and publishable after some edits and improvement.
Major Comments:
- Please comment on any evidence if available or provide speculation as to how the renal and extra-renal phenotypes caused by CEP164 mutations relate to the dysfunction of CEP164 in the centriole/cilia base versus in the nucleus?
- Do mutations in three DAP genes (CEP83, CEP164, and SCLT1) all result in complete deletion of proteins? Are there any cases of truncation or point mutations that produce mutant or truncated proteins that are still in the DAP complex but dysfunction?
I recommend some additional information in Table 1: 1) what type of human mutations and their impact on the protein product and its ability to interact, recruit, and structurally support; 2) a new column to describe cilia phenotype.
Minor Comments:
In Fig. 3, font size of labels needs to be increased to enhance visibility.
On Page 5, citations are needed for the following statements:
Distal appendages are required to remove the ciliogenesis inhibitor CP110 from the mother centriole by recruiting the Tau tubulin kinase 2 (Ttbk2).
Furthermore, the distal appendage protein CEP164 regulates ciliary-directed vesicular transport through its interaction with Rab8 and Rabin8.
Author Response
Responses to Reviewer 1:
Comments and Suggestions for Authors
This is a well-structured, comprehensive review of distal appendage proteins in nephronophthisis. This review not only summarizes recent advances but also will motivate further investigation of many significant unanswered questions about the mechanism and the role of DAPs in ciliogenesis. I found the current manuscript highly significant and publishable after some edits and improvement.
Response: Thank you for the positive evaluation of our manuscript.
Major Comments:
Comment No. 1
Please comment on any evidence if available or provide speculation as to how the renal and extra-renal phenotypes caused by CEP164 mutations relate to the dysfunction of CEP164 in the centriole/cilia base versus in the nucleus?
Response: Thank you for this interesting question to consider. After carefully reviewing the available literature, we did not identify any experimental data. Therefore, we added a paragraph speculating how the different subcellular functions of CEP164 may contribute to the pathogenesis.
Comment No. 2
Do mutations in three DAP genes (CEP83, CEP164, and SCLT1) all result in complete deletion of proteins? Are there any cases of truncation or point mutations that produce mutant or truncated proteins that are still in the DAP complex but dysfunction?
I recommend some additional information in Table 1: 1) what type of human mutations and their impact on the protein product and its ability to interact, recruit, and structurally support; 2) a new column to describe cilia phenotype.
Reply: We thank the reviewer for this important comment. We added to the manuscript and to Table 1 as requested.
Minor Comments:
Comment No. 1
In Fig. 3, font size of labels needs to be increased to enhance visibility.
Response: This was done.
On Page 5, citations are needed for the following statements:
Distal appendages are required to remove the ciliogenesis inhibitor CP110 from the mother centriole by recruiting the Tau tubulin kinase 2 (Ttbk2).
Furthermore, the distal appendage protein CEP164 regulates ciliary-directed vesicular transport through its interaction with Rab8 and Rabin8.
Response: Thank you for pointing this out. References were added.
Reviewer 2 Report
The review by Mansour and colleagues is well written, clearly organized and brings an updated view on the centrosome DAPs function in NPHP. Moreover, the figures are very explanatory.
I do not have any concerns about the manuscript which can be published in the present form.
Author Response
Responses to Reviewer 2:
Comments and Suggestions for Authors
The review by Mansour and colleagues is well written, clearly organized and brings an updated view on the centrosome DAPs function in NPHP. Moreover, the figures are very explanatory.
I do not have any concerns about the manuscript which can be published in the present form.
Response: Thank you for the positive evaluation of our manuscript.
Additional changes:
The manuscript was proofread by a native English speaker.